# Detection and Classification of Rice Infestation with Rice Leaf Folder (*Cnaphalocrocis medinalis*) Using Hyperspectral Imaging Techniques

Gui-Chou Liang [1], Yen-Chieh Ouyang [2] and Shu-Mei Dai [1,*]

1 Department of Entomology, National Chung Hsing University, Taichung 402, Taiwan; g107036008@mail.nchu.edu.tw
2 Department of Electrical Engineering, National Chung Hsing University, Taichung 402, Taiwan; ycouyang@nchu.edu.tw
* Correspondence: sdai5497@dragon.nchu.edu.tw; Tel.: +886-0963-234-136

**Abstract:** The detection of rice leaf folder (RLF) infestation usually depends on manual monitoring, and early infestations cannot be detected visually. To improve detection accuracy and reduce human error, we use push-broom hyperspectral sensors to scan rice images and use machine learning and deep neural learning methods to detect RLF-infested rice leaves. Different from traditional image processing methods, hyperspectral imaging data analysis is based on pixel-based classification and target recognition. Since the spectral information itself is a feature and can be considered a vector, deep learning neural networks do not need to use convolutional neural networks to extract features. To correctly detect the spectral image of rice leaves infested by RLF, we use the constrained energy minimization (CEM) method to suppress the background noise of the spectral image. A band selection method was utilized to reduce the computational energy consumption of using the full-band process, and six bands were selected as candidate bands. The following method is the band expansion process (BEP) method, which is utilized to expand the vector length to improve the problem of compressed spectral information for band selection. We use CEM and deep neural networks to detect defects in the spectral images of infected rice leaves and compare the performance of each in the full frequency band, frequency band selection, and frequency BEP. A total of 339 hyperspectral images were collected in this study; the results showed that six bands were sufficient for detecting early infestations of RLF, with a detection accuracy of 98% and a Dice similarity coefficient of 0.8, which provides advantages of commercialization of this field.

**Keywords:** rice; rice leaf folder; hyperspectral imaging; band selection; hyperspectral image classification; target detection

## 1. Introduction

Rice leaf folder (RLF), *Cnaphalocrocis medinalis* Guenée, is widely distributed in the rice-growing regions of humid tropical and temperate countries [1], and the developmental time of RLF decreases with an increase in temperature [2]. Due to global warming, RLF has become one of the most important insect pests of rice cultivation [3]. The larvae of RLF fold the leaves longitudinally and feed on the mesophyll tissue within the folded leaves. The feeding of RLF generates lineal white stripes (LWSs) in the early stage and then enlarge into ocher patches (OPs) and membranous OPs [4]. As the infestation of RLF increases, the number and area of OPs will increase. The feeding of RLF not only reduced the chlorophyll content and photosynthesis efficiency [4] but also provided a method for fungal and bacterial infection [5]. Therefore, the severe damage caused by RLF may cause 63–80% yield loss [6], and the highest record of the damaged area to rice cultivation in a single year exceeded 30,000 hectares [7].

The economic injury level of RLF, which is important for the determination of insecticide applications, has been established as 4.2% damaged leaves and 1.3 larvae per plant by the International Rice Research Institute [8]. However, it is laborious and time-consuming to visually inspect for damage. In addition, RLF is a long-distance migratory insect pest. The uncertain timing of the appearance of RLF means that farmers are unable to predict pest arrival, so to avoid damage by undetected infestations, farmers often preventively spray chemical insecticides, which generates unnecessary costs and environmental pollution [9,10].

Hyperspectral imaging (HSI) is a novel technique that combines the simultaneous advantages of imaging and spectroscopy and that has been investigated and applied in crop protection [11–15]. HSI, which contains spatial and spectral information, is given in Figure 1. The external damage and internal damage caused by pest infestations, such as yellowing/attenuation/defects and loss of pigments/photosynthetic activity/water content, respectively, can be identified by this system through image or spectral reflectance. Further automatic detection can be fulfilled by taking advantage of pest damage detection algorithms. For instance, constrained energy minimization (CEM) [16] and principal component analysis (PCA) [17] have been employed for band selection, and support vector machines (SVMs) [18], convolutional neural networks (CNNs) [19], and deep neural networks (DNNs) [20] are utilized for classification. Fan et al. [21] applied a visible/near-infrared hyperspectral imaging system to detect early invasion of rice streak insects. Using the successive projection algorithm (SPA) [22], PCA, and a back-propagation neural network (BPNN) [23] as classifiers to identify key wavelengths, the classification accuracy of the calibration and prediction sets was 95.65%. Chen et al. [24] also employed a visible/near-infrared hyperspectral imaging system to acquire images and further developed a hyperspectral insect damage detection algorithm (HIDDA) to detect pests in green coffee beans. The method combines CEM and SVM and achieves 95% accuracy and a 90% kappa coefficient. In addition, spectroscopy technology has been applied to detect plant diseases [25], the quality of agricultural products [26], and pesticide residues [27].

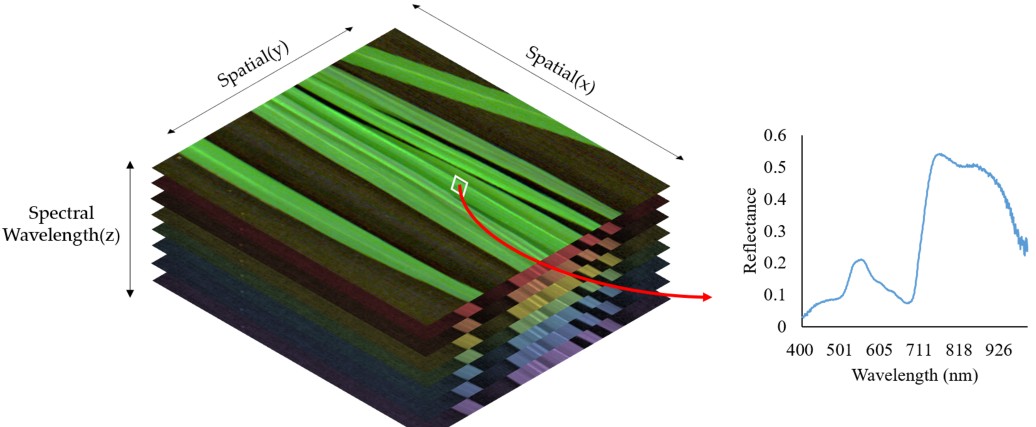

**Figure 1.** Two-dimensional projection of a hyperspectral cube.

To effectively manage RLF with a rational application of insecticides, an artificial-intelligent inspection of economic injury levels is necessary. The purpose of this study is to establish a model for detecting early infestation of RLF based on visible light hyperspectral data exploration techniques and deep learning technology. The specific objectives include (1) predefining the region of interest (ROI); (2) data preprocessing through a band selection and band expansion process (BEP); (3) simultaneously combining a deep learning network to train the model and to classify multiple different levels of damage; (4) using an automatic target generation program (ATGP) algorithm [28] to test unknown samples to fully automate the process and optimize the process to shorten the prediction time; and (5) es-

tablishing the spectral signatures of damaged leaves caused by RLF, which can serve as an expert system to provide valuable resources for the best timing of insecticide application.

## 2. Materials and Methods

### 2.1. Insect Breeding

The RLF in this study was collected from the Taichung District Agricultural Research and Extension Station. The larvae were raised in insect rearing cages (47.5 × 47.5 × 47.5 cm$^3$, MegaView Science Co., Ltd., Taichung, Taiwan) with corn seedlings (agricultural friend seedling Yumeizhen) and maintained at 27 ± 2 °C and 70% relative humidity during a photoperiod of 16:8 h (L:D). The adults were reared in a cage with 10% honey at 27 ± 2 °C and 90% relative humidity, which allows adults to lay more eggs.

### 2.2. Preparation of Rice Samples

The variety Tainan No. 11, which is the most prevalent cultivar planted in Taiwan, was selected for this study. Larvae were grown in a greenhouse to prevent the infestation of insect pests and diseases. To obtain different levels of damage caused by RLF, e.g., LWS and OP, 1st-, 2nd-, 3rd-, 4th-, or 5th-instar larvae of RLF were manually introduced to infest 40-day-old healthy rice for seven days, and three replicates were conducted for each treatment. Three different types of samples shown in Figure 2, e.g., healthy leaves (HL), LWS, and OP caused by RLF, were prepared for imaging acquisition and spectral information extraction.

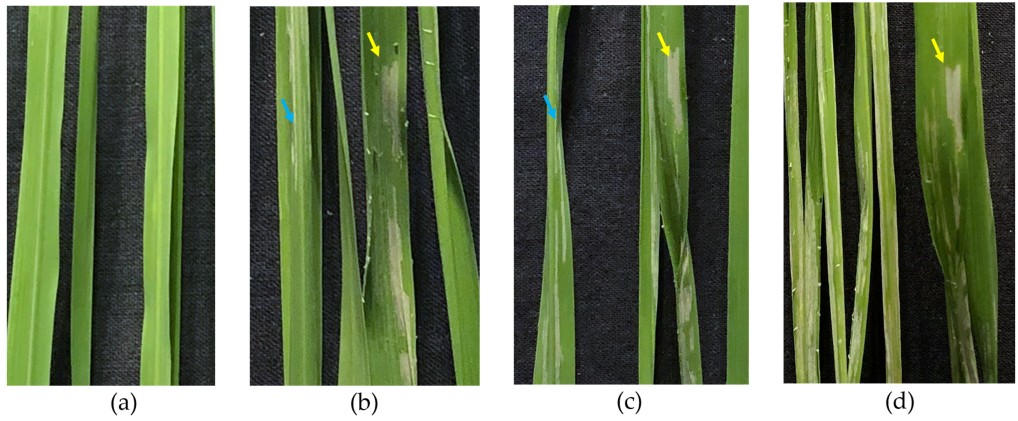

(a)       (b)       (c)       (d)

**Figure 2.** Appearance of healthy and damaged leaf types. (**a**) Healthy leaves, (**b**) lineal white stripe (LWS) caused by RLF (blue arrow) and LWS enlarge into ocher patch (OP) (yellow arrow) on Day 1 (D1), (**c**) LWS and OP on D2, and (**d**) OP on D6.

### 2.3. Hyperspectral Imaging System and Imaging Acquisition

#### 2.3.1. Hyperspectral Sensor

The hyperspectral scanning system employed in the experiment is shown in Figure 3. The hyperspectral image capturing system was composed of the following equipment: hyperspectral sensor, halogen light source, conveyor system, computer, and photographic darkroom isolated from external light sources. The hyperspectral sensor utilized in the study was a V10E-B1410CL sensor (IZUSU OPTICS), which contained visible and near-infrared (VNIR) bands with a spectral range from 380–1030 nm, a resolution of 5.5 nm, and 616 bands for imaging. The type of camera sensor is an Imspector Spectral Camera, SW ver 2.740. The halogen light sources used to illuminate the image were "3900e-ER", and the power was 150 W. Halogen lights were simultaneously illuminated on the left and right sides and focused on the conveyor track at an incident angle of 45 degrees to reduce shadow interference during the sampling process. The temperature and relative humidity in the laboratory were kept at 25 °C and 60%, respectively. A conveyor belt was designed to deliver rice plants for acquiring hyperspectral images by line scanning (Figure 3). Both the

speed of the conveyor belt and the halogen lights were controlled by computer software. The distance between the VNIR sensor and the rice sample was 0.6 m.

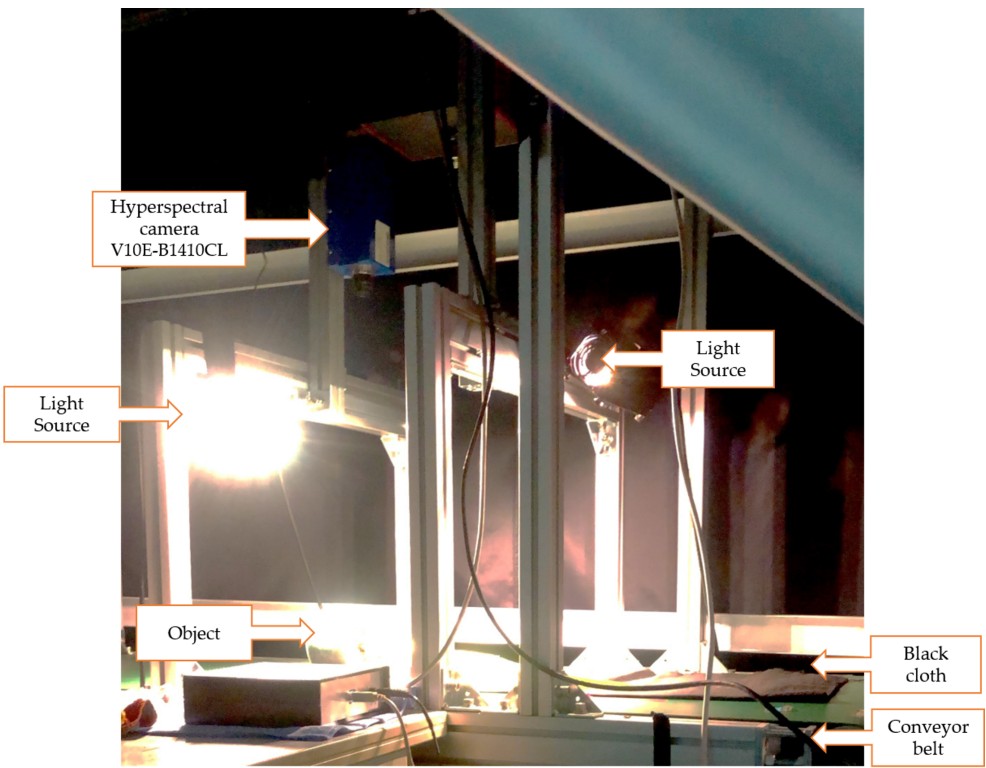

**Figure 3.** Hyperspectral imaging system.

### 2.3.2. Image Acquisition

The damage to leaves infested with different RLF larvae (from 1st to 5th instar larvae) for various durations of feeding (1–6 days) was assessed using VNIR hyperspectral imaging. Leaves were placed flat on the conveyor belt to scan the image at every 90° turn to enlarge the dataset. The exposure time for scanning was 5.495 ms, and the number of pixels in each scan raw was 816. Healthy leaves without RLF infestation were selected as the control. Before taking the VNIR hyperspectral images, light correction was conducted, and all processing of images was conducted in a dark box to avoid interference from other light sources. In total, 339 images, including 52 images of healthy leaves and 69, 32, 48, 52, 52, and 34 images of leaves infested for 1 day to 6 days, were taken.

### 2.3.3. Calibration

To eliminate the impacts of uneven illumination and dark current noise, the object scan, reference dark value, and reference white value are needed to perform the normalization step. To reduce noise and avoid the influence of dark noise, the original hyperspectral image must be calibrated according to the following formula [21]:

$$R_C = \frac{R_0 - B}{W - B} \tag{1}$$

where $R_0$ is the raw hyperspectral image, $R_C$ is the hyperspectral image after calibration, $W$ is the standard white reference value with a Teflon rectangular bar, and $B$ is the standard black reference value obtained by covering the lens with a lens cap.

### *2.4. Spectral Information Extraction*

Removing the background of the image will help extract useful spectral information and reduce noise. The background removal process performs binary segmentation

through the Otsu method, dividing the image into background and meaningful parts with similar features and attributes [29], including healthy, RLF-infested, and other defective leaves. To reduce unnecessary analysis work, the first step is to separate plant pixels from non-plant pixels. This task directly converts the grayscale image from the true-color image or generates a single channel image (grayscale image) based on a simple index (e.g., Excess Green [30]). Second, the threshold value is obtained using the Otsu method; the grayscale value of each pixel point is compared with the threshold value, and the pixel is classified as a target or background based on the result of the comparison [31]. Since plants and backgrounds have very different characteristics, they can be separated quickly and accurately.

Third, the images that had been removed from the background were applied to determine the ROI using the CEM algorithm [16]. CEM has been widely employed for target detection in hyperspectral remote sensing imagery. CEM detects the desired target signal source by using a unity constraint while suppressing noise and unknown signal sources; it also minimizes the average energy of output. This algorithm generates a finite impulse response filter through a given vector as the d value to suppress regions that are not related to the features of the ROI. The vector indicates the spectral reflectance of a pixel in this study, and the ROI was predefined as an RLF-infested region in the images of rice leaves, e.g., Figure 2b,c. The results of the CEM processing of the image show the enhanced characteristics of pixels similar to the target feature d value. Using the Otsu method, if the pixel value exceeds the threshold, the feature similarity is set to 1; otherwise, it is set to 0. Last, a binary image is obtained. This algorithm is an efficient method of pixel-based detection [32].

### 2.5. Band Selection

Since HSIs usually contain hundreds of spectral bands, full-band analysis of the spectrum is not only time-consuming but also too redundant. To decrease the analysis time and redundancy, the first step of data analysis is to determine the key wavelengths. The way to achieve this goal is to select highly correlated wavelengths by comparing the reflectance and to maximize the representativeness of the information by decorrelation. Various band spectral methods based on certain statistical criteria have been selected to achieve this purpose [33]. The concept of band selection is similar to feature extraction in image processing, which can improve the accuracy of identification and classification.

### 2.5.1. Band Prioritization

In the band prioritization (BP) part, the priority of the spectral bands will be calculated by statistical criteria [27]. Five criteria—variance, entropy, skewness, kurtosis, and signal-to-noise ratio (SNR)—were chosen to calculate the priority of the spectral bands in this work. Thus, each spectral band has a priority and can be ranked with high priority.

### 2.5.2. Band Decorrelation

When applying BP in the band selection process, the correlation between each band will highly affect the priority score. Neighboring bands will frequently be selected because of the high correlation between each band. Nevertheless, these redundant spectral bands are not helpful for improving detection performance. Therefore, to solve this problem, band decorrelation (BD) is utilized to remove these redundant spectral bands.

In this study, spectral information divergence (SID) [34] was applied for BD and utilized to measure the similarity between two vectors. By calculating the SID value, a threshold will be set to remove the bands with high similarity. The formula is:

$$k(b_i, b_j) = D(b_i \parallel b_j) + D(b_j \parallel b_i) \tag{2}$$

The parameter "*b*" represents a vector of spectral information, and $D(b_i \parallel b_j)$ denotes Kullback–Leibler divergence, that is, the average amount of difference between the self-information of $b_j$ and the self-information of $b_i$, and vice versa.

### 2.6. Band Expansion Process

Although the band selection-acquired spectral images can reduce storage space and processing time, some of the original features of the spectra were lost. To solve the problem of information loss after band selection, the difference in reflectivity can be increased by expanding the band to increase the divergence. The concept of the BEP [35] is derived from the fact that a second-order random process is generally specified by its first-order and second-order statistics. These correlated multispectral images provide missing but useful second-order statistical information about the original hyperspectral images. The second-order statistical information utilized for the BEP includes autocorrelation, cross-correlation, and nonlinear correlation to create nonlinearly correlated images. The concept of generating second-order, correlated band images coincide with the concept of covariance functions employed in signal processing to generate random processes. Even though there may be no real physical inference for the band expansion process, it does provide an important advantage for addressing the problem of an insufficient number of spectral bands.

### 2.7. Data Training Models

Hyperspectral imaging data analysis is based on pixel-based classification and target recognition, using low-level features (such as spectral reflectance and texture) as the bedding, and the output feature representation at the top of the network can be directly input to subsequent classifiers for pixel-based classification [36]. The classification of this pixel is particularly suitable for deep learning algorithms to learn representative and discriminative features from the data in a hierarchical manner. In this study, the input neuron is the reflectance of a pixel. The input layer has 466 neurons in the full band, 6 neurons after band selection, and 27 neurons after band expansion. As shown in Figure 4a,b, the reflectivity of the HL, D1 OP, and D6 OP samples was divided into three categories. The model is trained with four hidden layers, and the learning rate parameter is 0.001. A softmax classifier was provided in the DNN terminal, and the classification results of the spectrum were obtained. The classified result was compared with the ground truth to calculate the accuracy. The model repeated the cross-validation ten times and averaged it as its overall accuracy (OA).

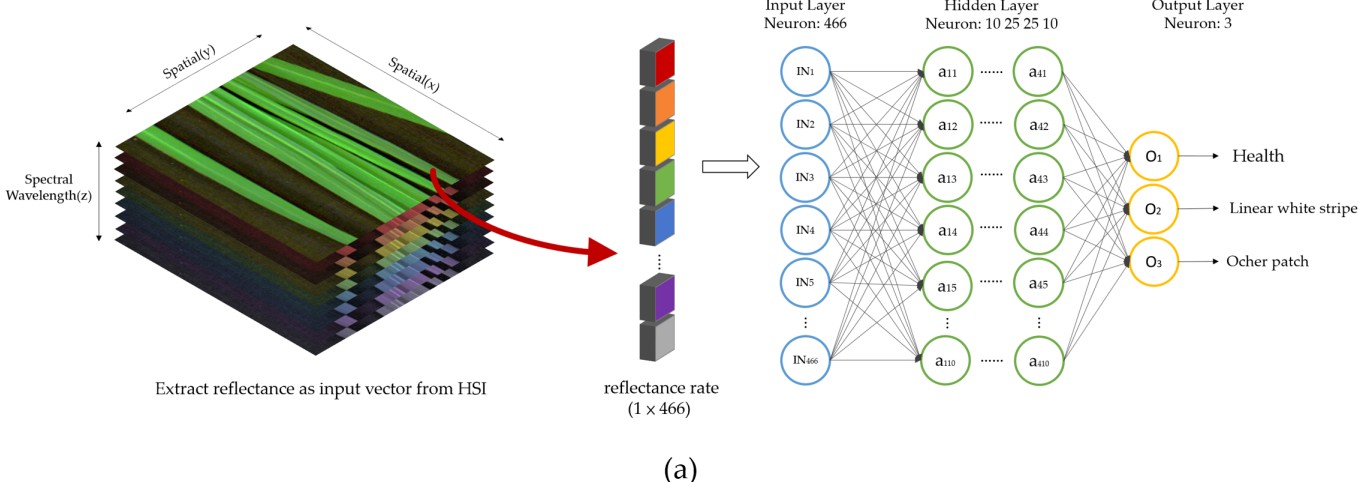

(a)

**Figure 4.** *Cont.*

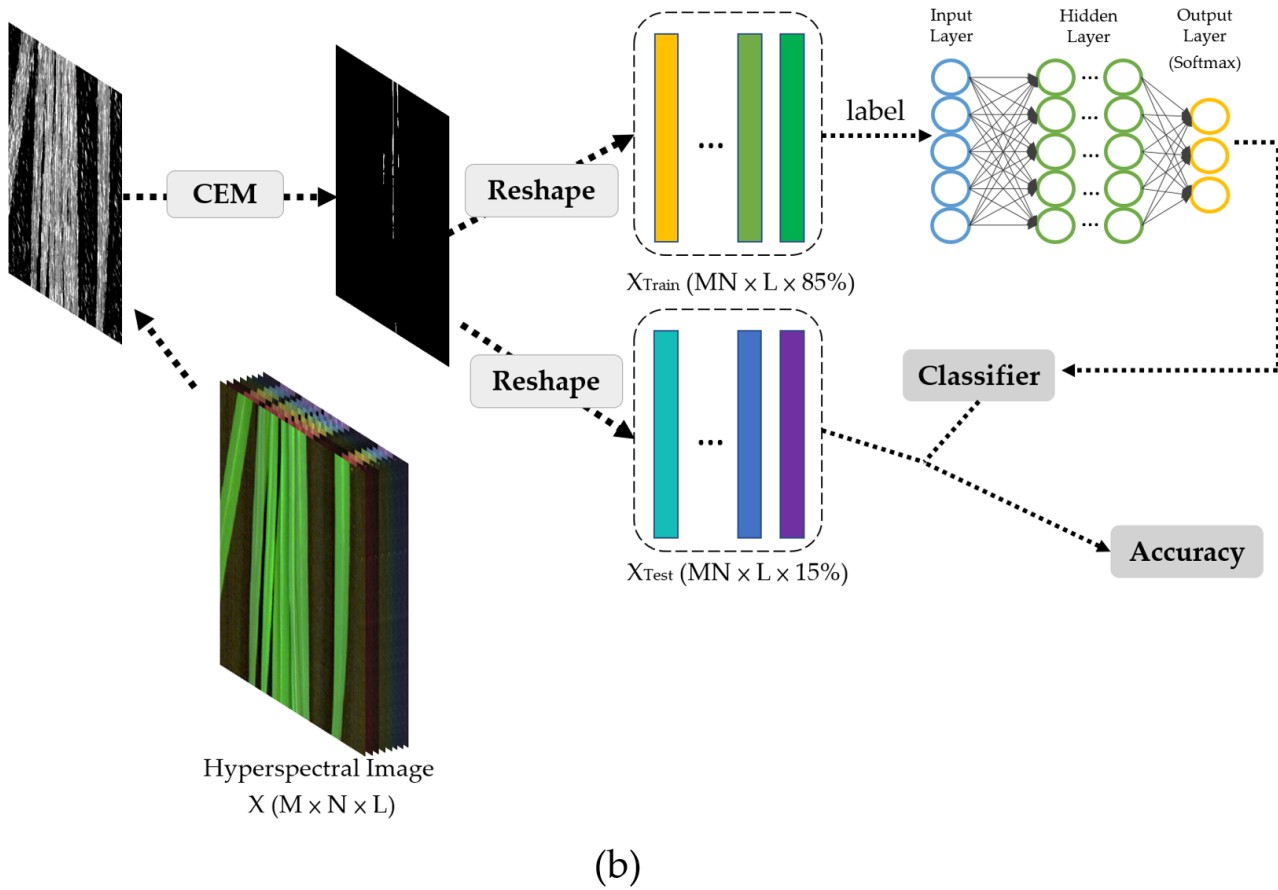

(b)

**Figure 4.** (**a**) DNN model architecture. (**b**) Flowchart of classifying reflectance using DNN.

Figure 5 depicts the data training flowchart of this study, starting with hyperspectral image capture. First, the reflectivity is extracted from the ROI as a ground truth, which was selected by the entomologist. Second, the reflectance dataset applied in the full-band spectrum was processed in the same way to build DNN models after band selection. Last, the band selection dataset was processed by the BEP to build a DNN model.

The DNN model is constructed using three processes: full bands, band selection, and BEP. Each classification model has the best weight evaluated by its own model. Three DNN classification process models are constructed based on randomly distributed datasets, including 70% training, 15% validation, and 15% testing (as shown in Table 1). In the testing phase, the accuracy of each classification situation will be compared, and the OA of multiple classifications will be integrated. As a result, the most suitable model for identifying the classification was obtained.

**Table 1.** Number of pixels used for band section, training, and testing in the rice dataset.

| | Sample Types | | |
|---|---|---|---|
| | **HL** | **D1 OP** | **D6 OP** |
| Band selection [1] | 297 | 301 | |
| Pixel numbers used for DNN Training | 5936 | 6015 | 6962 |
| Pixel numbers used for DNN Testing | 1000 | 1000 | 1000 |

[1] Band selection number = 5% of training number.

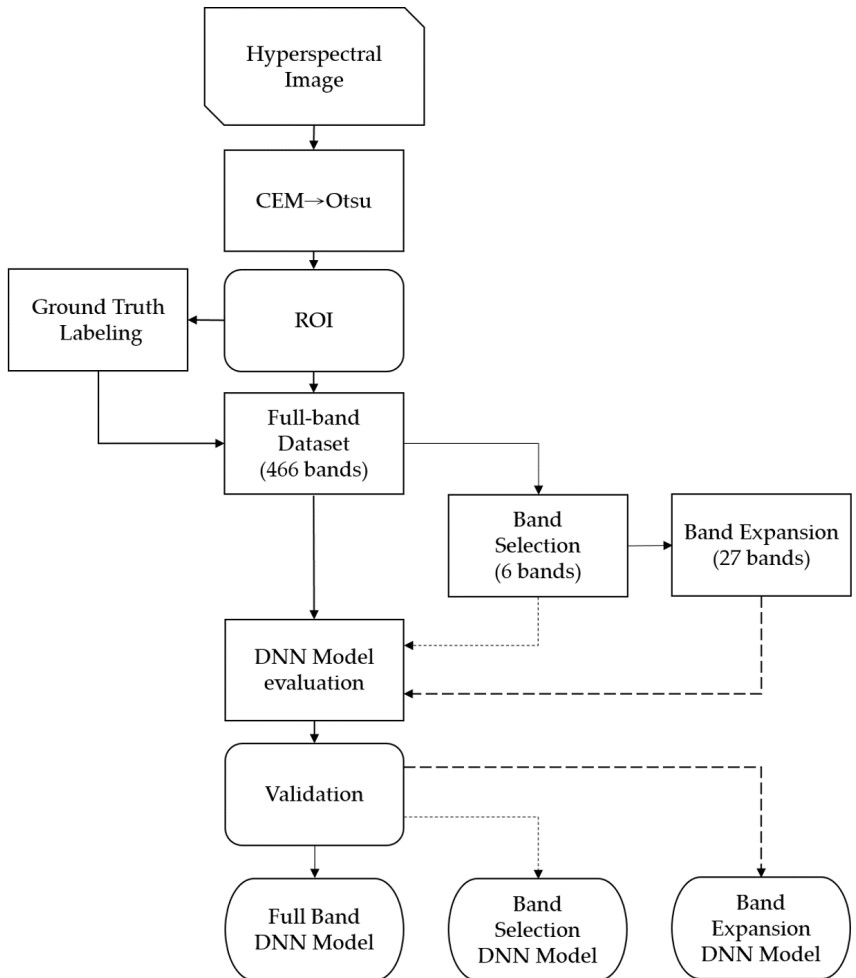

**Figure 5.** Data training flowchart of full bands, band selection, and band expansion process.

*2.8. Model Test for Unknown Samples*

To apply the spectral reflectance of unknown samples of healthy leaves, early and late OPs leave machine learning. The first step is to quickly determine the ROI to reduce the time required for image recognition. To achieve this goal, a method that combines an ATGP [28] and CEM is proposed. The ATGP is an unsupervised target recognition method that uses the concept of orthogonal subspace projection (OSP) to find a distinct feature without a priori knowledge. The ATGP method was employed to identify the target pixel in the hyperspectral image, and all the similar pixel data obtained were averaged as the d value of CEM.

Figure 6a,b shows the flow chart of the unknown sample prediction model. To automate the detection process, first, the full-band HSI, band selection, and BEP of the rice sample must be calibrated. Second, through the combined method of the ATGP and CEM, the Otsu method is utilized to mark the ROI. The ROI obtained from the full band, band selection, and BEP is classified by the corresponding DNN model and is labeled HL, early OP, or late OP by entomologists according to the occurrence of damage caused by RLF. The labeled ROI will be utilized to verify the prediction results of the DNN model. Five analysis methods, such as CEM_Full-band→DNN_Full-band, CEM_band selection→DNN_band selection, CEM_band selection→DNN_BEP, CEM_BEP→DNN_band selection, and CEM_BEP→DNN_BEP, are established to evaluate the prediction performance.

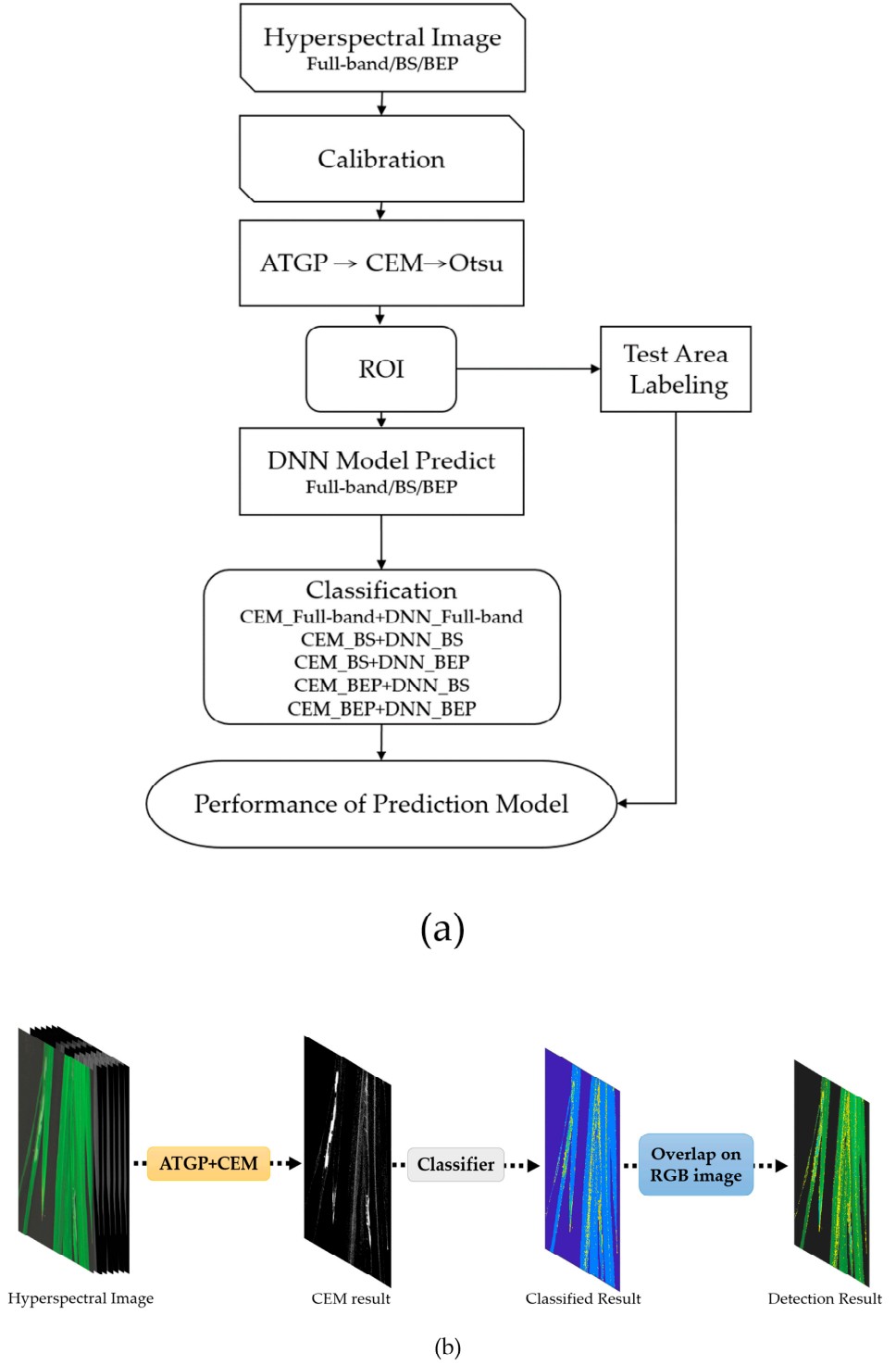

**Figure 6.** (**a**) Flow chart of the DNN model is used to predict unknown samples. (**b**) Flow chart of DNN model prediction.

Last, the model classification results were visualized and overlaid on the original true-color images, and agricultural experts verified the actual situation afterward to compare the performance of the models.

### 2.9. Predict Unknown Samplings

After a cross-validated predictive model has been established, a completely unknown sample with different data from the training set is needed to test its robustness. Eligible

samples were obtained from the field. To fix other conditions, the retrieved samples were also photographed with a push-broom hyperspectral camera.

Many different evaluation metrics have been mentioned in the literature. The confusion matrix [37] was selected as a measure of model accuracy. A true positive (TP) is a correct detection of the ground truth. A false positive (FP) is an object that is mistaken as true. A false negative (FN) is an object that is not detected, although it is positive.

However, it is not enough to rely on the confusion matrix alone. An additional pipeline of common evaluation metrics was needed to facilitate a better comparison of classification models. The following metrics were employed for the evaluation in this study:

(i)    recall

$$\text{Recall} = \frac{\text{TP}}{\text{TP} + \text{FN}} \tag{3}$$

(ii)   precision

$$\text{Precision} = \frac{\text{TP}}{\text{TP} + \text{FP}} \tag{4}$$

(iii)  Dice similarity coefficient

$$\text{Dice similarity coefficient} = 2 \times \frac{\text{TP}}{(2 \times \text{TP} + \text{FP} + \text{FN})} \tag{5}$$

The recall is the ability of the model to detect all relevant objects, i.e., the ability of the model to detect all detected bounding boxes of the validation set. Precision is the ability of the model to identify only relevant objects. The Dice similarity coefficient (DSC) is an ensemble similarity measure function that is usually applied to calculate the similarity between two samples in the value range between 0 when there is no overlap and 1 when there is complete overlap.

## 3. Results and Discussion

### 3.1. Images and Spectral Signatures of Healthy and RLF-Infested Rice Leaves

When larvae of RLF feed on rice leaves, they generate LWS or OP on the leaves. As time passes, the LWSs are enlarged into a patch; the color of the patch gradually turns from white to ocher; and the images and spectral signatures of these patches also change during this process, as shown in Figure 7a,b, respectively. The spectral signatures of HL and OP in Figure 7b were obtained manually, according to entomological experts. The OPs have higher reflectance than HL in the blue to red wavelength range. Among these spectral bands, the longer the infestation period is, the higher the reflectance, e.g., day 6 (D6) > D5 > D2 > D1. However, only the reflectance of D6 OP is higher than that of HL at the NIR wavelength (Figure 7b). The reflectance of D1-OP is much lower than the HL reflectance, and the reflectance of D2- and D5-OPs is approximately the same as that of healthy leaves. The decrease in reflectance in D1 OP at NIR was mainly due to the destruction of leaf structure, which caused photon scattering [38]. These results suggest that the early defects caused by RLF have very different spectral signatures of vectors from the subsequent damage of infestation. Differences in the spectral properties between the early phase of damage and the late phase of damage, which could serve as a basis for the early identification of RLF infestations.

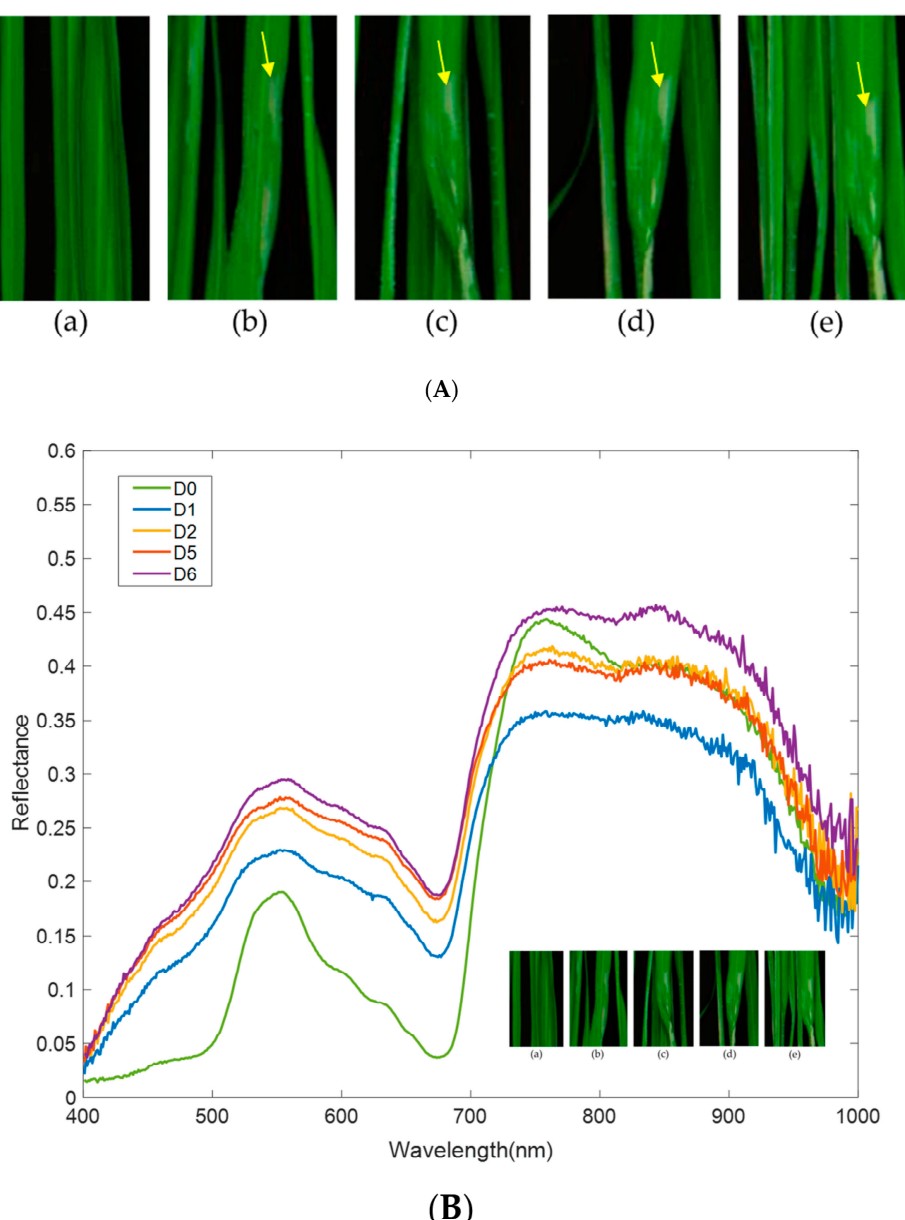

**Figure 7.** (**A**) Hyperspectral images of healthy leaves on day 0 (**a**) and ocher patches (yellow arrow) infested by rice leaf folders on day 1 (**b**), day 2 (**c**), day 5 (**d**), and day 6 (**e**). (**B**) Spectral signature and corresponding hyperspectral images of the healthy leaves (D0) and ocher patches (from D1 to D6) caused by RLF.

### *3.2. Band Selection and Band Expansion Process*

The HSI and spectral signature from the full band system shown in Figure 7a,b contain considerable redundant information that slow the analysis efficiency and consume too much storage space. Therefore, band selection and the BEP were employed to select the most informative bands to increase the analysis efficiency and reduce storage space. To more effectively detect early RLF infection, the number of training sessions for HL and D1 OP was 5%, as shown in Table 1; these sessions were chosen to perform band selection. Five criteria were utilized in BP to calculate the priority of each band from the full-band signature of HL and D1 OP, and then, a value of 2.5 for SID was chosen as the threshold for BD to remove the adjacent bands with high similarity for D1 OP. Six bands of 489, 501, 603, 664, 684, and 705 nm, which had the largest difference in reflectance between HL and D1 OP, were selected as candidates through BP and BD using the criteria of entropy (Figure 8a,b).

To adapt to the cheaper and easy-to-use, six-band handheld spectrum sensor, we only chose the six-band spectrum. The results of band selection using the other four criteria are shown in Supplementary Table S1 and Figure S1. Furthermore, the six bands were expanded to 27 bands using the BEP to improve the deficiency caused by band selection.

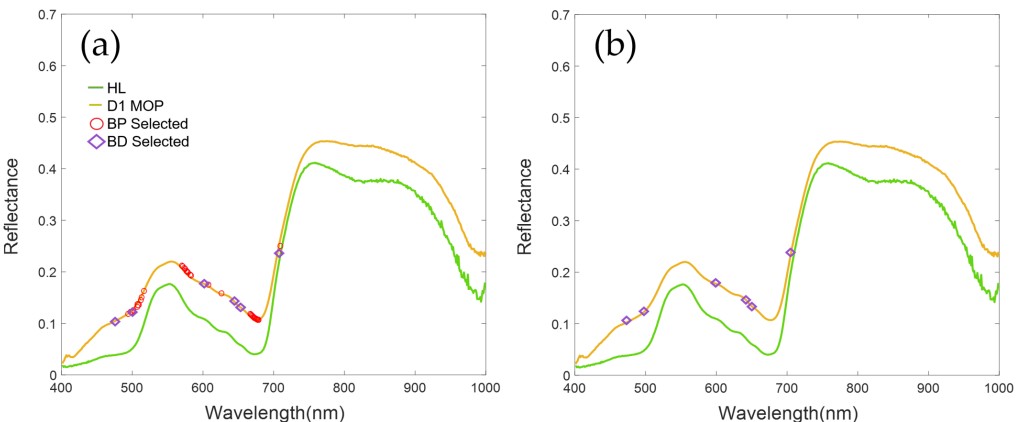

**Figure 8.** Bands selected through band prioritization (**a**) and band decorrelation (**b**) using criteria of entropy. Red circles denote bands selected from band prioritization, and purple diamonds denote bands selected after band decorrelation.

### 3.3. ROI Detection with CEM in Full Bands, Band Selection, and Band Expansion Process

CEM, a standard linear detector, was selected as a filter in this study to quickly identify the ROI. CEM increases the accuracy of automated detection and reduces the analysis time. The spectral signature of the OP that appeared on D1 in Figure 7b was employed as the d value of CEM to detect damaged leaves caused by RLF. Figure 9 shows the effect of different degrees of enhancement on ROI detection in the case of the full band, band selection, and BEP and the results of k-means clustering as a contrast. In the case of full bands, very minimal damage caused by RLF was detected (Figure 9b). The abundance of spectral data increases the complexity of detection and reduces the spectral reflectance resulting from RLF. On the other hand, the ROI detection in the cases of band selection reveals almost all the damage shown in Figure 9a. This finding indicates that band selection can achieve the best performance in ROI detection through CEM (Figure 9c). In the case of the BEP, the result of CEM is better than the full bands but not as good as band selection (Figure 9d).

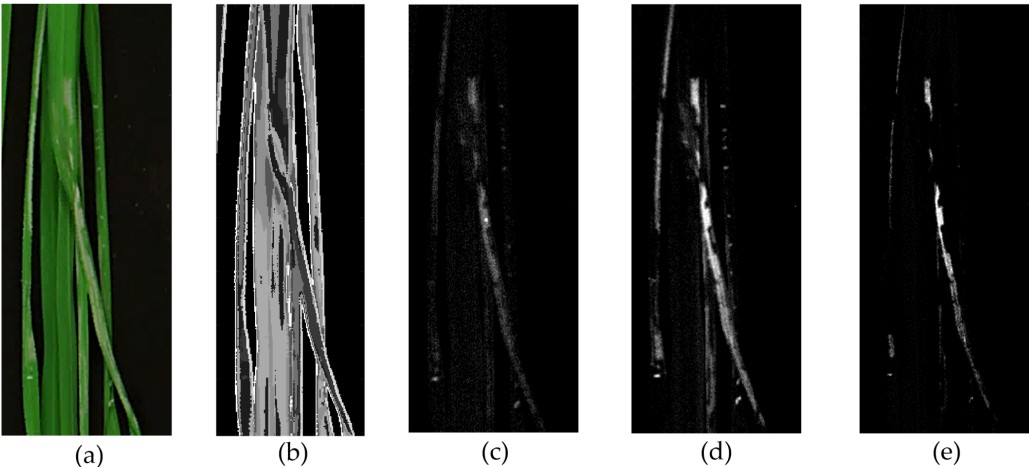

**Figure 9.** Region of interest detection with k-means (k = 10) or constrained energy minimization algorithm on different datasets using the reflectance of the D1 ocher patch as a d value on rice leaves. (**a**) True-color image, (**b**) k-means in full bands, (**c**) CEM in full bands, (**d**) band selection, and (**e**) band expansion process.

### 3.4. DNN Model for Classification of Testing Dataset

The DNN multilayer perceptron model is suited for HSI data for classification because the spectral reflection of each pixel can form a vector. Even if we have fewer images, we can still use enough pixels as samples for analysis. Therefore, this study does not require thousands of images to train a set of deep learning models, which greatly reduces the tedious work of collecting samples and the difficulty of controlling sample conditions.

Table 2 describes the results of the OA verification using the DNN models of the full bands, band selection (6 bands), and BEP (27 bands). The confusion matrix [37] was utilized to evaluate the classification performance; the complete confusion matrix calculated for DNN classification is shown in Supplementary Figure S2. In the case of full bands, the OA (95%) and performance are the best in the classification of various situations, but a longer time (14.88 s) is needed than band selection and BEP in classification. The application of band selection saves approximately half the time of full bands, but it will also reduce the classification accuracy. Except for HL, the accuracy of early and late OPs decreased after band selection, which may be attributed to a decrease in some spectral information. The accuracy of the BEP is not higher than that of band selection, as expected, and it is possible that BEP amplifies the noise and interferes with the classification ability. Among the five criteria, the OA of classification is the best among the bands selected by entropy. In terms of entropy, the accuracy of early OP from band selection is approximately 4% higher than that from BEP.

**Table 2.** Results for the testing dataset for DNN classification in different bands. The best performance is highlighted in red.

| Model | Criteria | Accuracy (%) | | | OA [3] (%) | Time (s) |
|---|---|---|---|---|---|---|
| | | HL | Early [1] OP | Late [2] OP | | |
| Full-band | - | 97.3 | 93.6 | 94.0 | 95.0 | 14.88 |
| Band selection (6 bands) | Variance | 96.0 | 84.4 | 85.1 | 88.5 | 7.18 |
| | Entropy | 97.2 | 87.1 | 86.5 | 90.3 | 5.79 |
| | Skewness | 95.7 | 82.5 | 81.6 | 86.6 | 4.96 |
| | Kurtosis | 97.4 | 78.7 | 86.5 | 87.5 | 6.32 |
| | SNR | 97.8 | 78.4 | 78.9 | 85.0 | 6.98 |
| Band expansion process (27 bands) | Variance | 97.0 | 83.3 | 84.1 | 88.1 | 7.88 |
| | Entropy | 97.1 | 82.8 | 83.5 | 87.8 | 6.83 |
| | Skewness | 96.6 | 76.2 | 81.7 | 84.8 | 5.85 |
| | Kurtosis | 96.3 | 78.2 | 86.8 | 87.1 | 6.79 |
| | SNR | 96.9 | 78.0 | 81.3 | 85.4 | 7.43 |

[1] Early OP comprises a set of D1 and D2 OP. [2] Late OP comprises a set of D5 and D6 OP. [3] OA is an abbreviation for overall accuracy.

### 3.5. Prediction of Unknown Samples

The predictions were carried out using ROIs obtained from full bands, band selection, and the BEP, as shown in Figure 6. CEM was applied to suppress the background and to detect the ROI. The DNN models of full bands, band selection, and the BEP were used as classifiers to predict unknown samples through five analysis methods. For band selection and the BEP, bands selected by entropy were selected as examples according to the results of Table 2 to execute the prediction. Figure 10 shows the results of the true-color image (a), ground truth (b), and predictions from an unknown sample (c–g). The ground truth was determined by entomologists and given different colors to distinguish HL (green) from OP (red). Figure 10c–g shows the classification results from the full bands and band selection/BEP, respectively, which were also colored for visualization. Figure 10d,e shows the best results as expected, in which the predicted areas of the ROI were approximately the same as the ground truth (Figure 10). However, the predicted ROI in Figure 10c was distributed over the rice leaves in addition to the ROI of the ground truth.

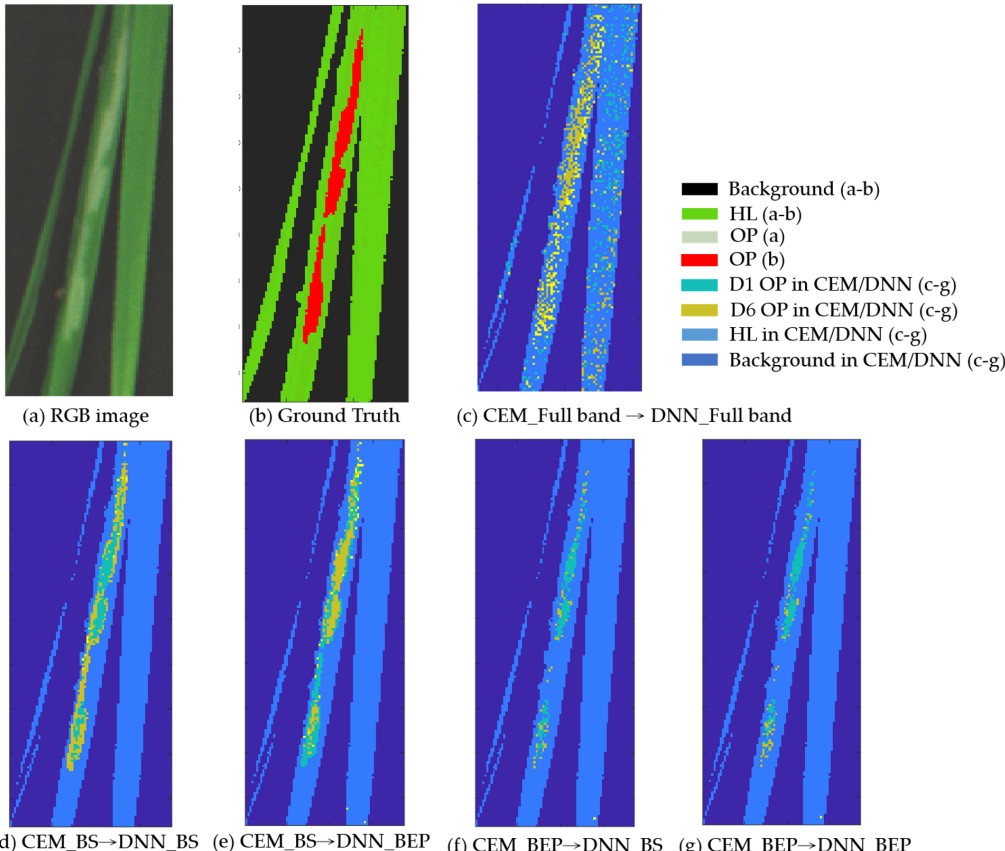

**Figure 10.** Prediction of spectral information from unknown rice sample: (**a**) true-color image, (**b**) Ground Truth, (**c**) CEM_Full-band→DNN_Full-band, (**d**) CEM_band selection→DNN_band selection, (**e**) CEM_band selection→DNN_BEP, (**f**) CEM_BEP→DNN_band selection, and (**g**) CEM_BEP→DNN_BEP.

The performance of the pixel classification of DNN models was verified by comparing the prediction results with ground truth using a confusion matrix; the results are shown in Tables 3 and 4. Similar to the results in Figure 10, the analysis methods show that CEM_band selection→DNN_band selection showed the best prediction performance (Table 3) because this method showed the highest TP (correct identification of OP) and overall accuracy (OA) and the lowest FN (misidentification of OP). However, very high false positives (FPs) were obtained from the methods of CEM_Full-band→DNN_Full-band, CEM_band selection→DNN_band selection, and CEM_band selection→DNN_BEP (Figure 10c–e). The high FP value of CEM_Full-band→DNN_Full-band may be derived from the scattered distribution of predicted ROI, while the high FP values of CEM_band selection→DNN_band selection and CEM_band selection→DNN_BEP predicted area of ROI may be derived from the predicted areas of ROI that are undetectable by the naked eye. To prove the above observation, the images of Figure 10d or Figure 10e were overlaid with ground truth (Figure 10b). The extra predicted area around the ROI of ground truth in Figure 11d,e should be the early infestation of RLF that cannot be detected by human eyes.

To verify the necessity of using CEM to extract ROI, the DNN classification results of the background-removed images are shown in Supplementary Table S2 and Figure S3. The results show that the accuracy of DNN classification after CEM processing is approximately 22% higher than that of the DNN applied directly to remove the background.

**Table 3.** Accuracy of DNN classification evaluated by the confusion matrix.

| Analysis Method | Pixel Number | | | | OA (%) |
|---|---|---|---|---|---|
| | TP [2] | FP [3] | TN [4] | FN [5] | |
| CEM_Full-band→DNN_Full-band | 317 | 341 | 11,781 | 289 | 95.05 |
| CEM_band selection→DNN_band selection [1] | 497 | 138 | 11,984 | 109 | 98.05 |
| CEM_band selection→DNN_BEP | 488 | 138 | 11,984 | 178 | 97.98 |
| CEM_BEP→DNN_band selection | 318 | 17 | 12,105 | 288 | 97.60 |
| CEM_BE→DNN_BEP | 302 | 18 | 12,104 | 304 | 97.47 |
| | Positive [6] | | Negative [7] | | |
| Ground Truth | 606 | | 12,122 | | |

[1] Bands selected by Entropy. [2] TP represents the correct identification of OP; [3] FP denotes the health misidentification of HL; [4] TN indicates the correct identification of health HL; [5] FN represents misidentification of OP; [6] OP is positive, and [7] non-OP is negative.

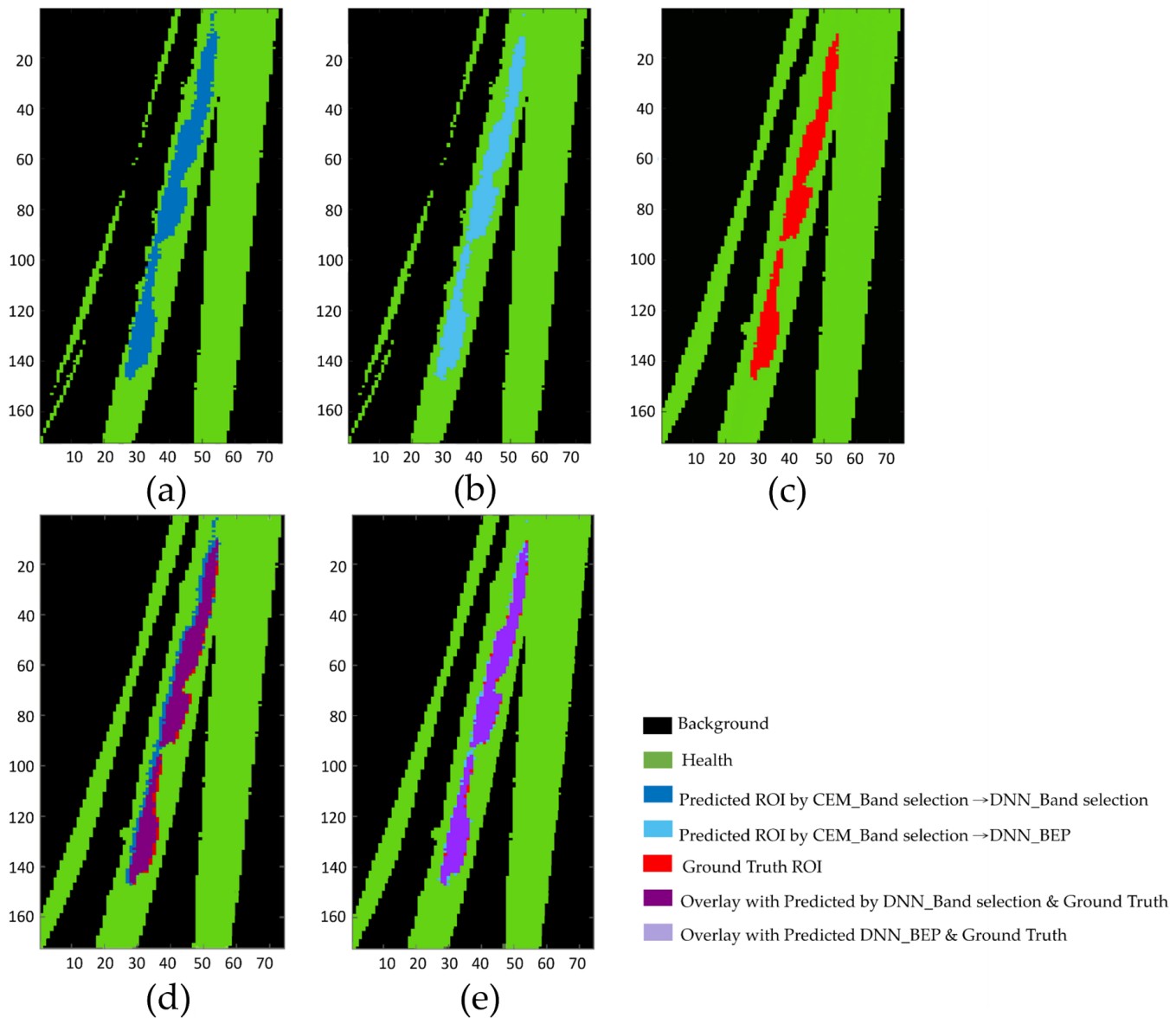

**Figure 11.** Overlaid images of the predicted ROI with the ground-truth ROI for evaluating the performance of DNN classification. (**a**) Predicted ROI with CEM_band selection→DNN_band selection, (**b**) predicted ROI with CEM_band selection→DNN_BEP, (**c**) Ground Truth, (**d**) Overlay with (**a,c,e**) Overlay with (**b,c**).

The performance of DNN classification was further evaluated by the metrics of recall, precision, accuracy, and DSC, as shown in Table 4. The analysis method of CEM_band selection→DNN_band selection was again rated as the best model for predicting unknown samples, as it had the highest accuracy, recall, and DSC and took the shortest time. Although the analysis method of CEM_band selection→DNN_BEP also showed reasonably good performance, the overall results indicated that six bands obtained from band selection are good enough to detect the early OP caused by RLF. The analysis method of CEM_BEP→DNN_band selection has the highest precision, but its recall and DSC are lower than those of CEM_band selection→DNN_band selection and CEM_band selection→DNN_BEP.

**Table 4.** Evaluation metrics of DNN prediction. The best performance is highlighted in red color.

| Analysis Method | Recall | Precision | Accuracy | Dice Similarity Coefficient | Time (s) |
|---|---|---|---|---|---|
| CEM _Full-band→DNN_Full-band | 0.523 | 0.482 | 0.951 | 0.670 | 3.672 |
| CEM_band selection→DNN_band selection | 0.820 | 0.783 | 0.981 | 0.801 | 0.336 |
| CEM_band selection→DNN_BEP | 0.805 | 0.780 | 0.980 | 0.755 | 0.381 |
| CEM_BEP→DNN_band selection | 0.525 | 0.949 | 0.976 | 0.676 | 0.559 |
| CEM_BEP→DNN_BEP | 0.498 | 0.915 | 0.974 | 0.652 | 0.604 |

Taking the OP as an example, the pixels of the ROI were utilized for prediction evaluation, and a confusion matrix was employed for performance in this study. As shown in Table 4, all analysis methods were successful in classification, and their accuracies reached at least 95%. The area of the block classified as OP is smaller than the actual situation, which is the case in Figure 11f. As shown in Figure 11d, CEM_band selection→DNN_band selection, the distribution of false positives was observed around the OP, which means that the earlier defects caused by insect pests could be identified as false-positive areas in hyperspectral images but could not be recognized in true-color images or human eyes.

*3.6. Discussion*

Automatic detection of plant pests is extremely useful because it reduces the tedious work of monitoring large paddy fields and detects the damage caused by RLF at the early stage of pest development and eventually stop further plant degradation. This study proposes an automatic detection method that combines CEM and the ATGP. CEM is an efficient hyperspectral detection algorithm that can efficiently handle subpixel detection [39]. The quality of the CEM results is determined by the d-value used as a reference. Therefore, it is important to provide a plausible spectral feature. The ATGP was applied to identify the most representative feature vector as the d-value from an unknown sample. Another problem with the CEM is that it only provides a rough detection result. The DNN was selected to classify the reflectance of the ATGP→CEM detection results. In addition, band selection and the BEP were chosen to identify the key wavelengths among the five criteria to save time and improve accuracy. The accuracy of CEM_band selection→DNN_band selection in predicting the performance of unknown samples reached 98.1%. Traditional classifiers such as linear SVM (support vector machine) and LR (logistic regression) can be attributed to single-layer classifiers, while decision trees or SVM with kernels are considered to have two layers [40,41]. However, deep neural architectures with more layers can potentially extract abstract and invariant features for better image or signal classification [42]. Our previous studies to detect Fusarium wilt on Phalaenopsis have shown this result [43]. In addition, we have used the Entoscan Plant imaging system to detect the infestation of RLF, but this system only covers 16 bands (390, 410, 450, 475, 520, 560, 625, 650, 730, 770, 840, 860, 880, 900, 930, and 960 nm) to obtain the Normalized Difference Vegetation Index. The results are shown in Supplementary Figure S4. It may not be specific enough to distinguish the damage caused by different pests. Therefore, we attempt to find a more representative vector from the spectral fingerprint of the hyperspectral imaging system to detect the infestation of RLF. At the same time, the band selection was used to remove redundant

information to achieve the time required for the automatic detection process. It is not only reducing the time by 2.45 times (from 8′11″ to 3′20″) but also reach higher accuracy (0.981) than that (0.951) in the full band. The time required for each stage of the prediction process is shown in Supplementary Figure S5. The six bands (489, 501, 603, 664, 684, and 705 nm) obtained through band selection are more representative than bands supplied by the Entoscan Plant imaging system and can be applied to the multispectral sensor of UAVs and portable instruments for field use. The methods, algorithms, and models we established in this paper will be applied to other important rice insect pests and verified in the field by using either UAVs or portable instruments that carry the multispectral sensor. In addition, a platform to integrate all this information will be established to interact with farmers.

Other studies [44,45] used conventional true-color images, which can only classify spatial information based on their color and shape and identify damage that is clearly visible by the naked eye. Compared with previous studies, the DNN was based on high spectral sensors to provide spectral information, which can detect pixel-level targets and retain the spatial information of the original image. The authors [44,45] employed the CNN to detect pests and achieved a classification accuracy of 90.9% and 97.12%, respectively. The method proposed in this paper is slightly higher than the final accuracy of CNN. Although it can simultaneously classify multiple insect pests and diseases, it often causes confusion. In addition, their studies were conducted with images of the late damage stage and could not classify the level of infestation. In addition, most image classifications are trained by a CNN. CNNs often need to collect a large number of training samples, and it is difficult to obtain a large number of sufficient training images in a short period of time. In contrast, hyperspectral image classification based on spectral pixels can be trained by a DNN, which means that even a single hyperspectral image can have a large amount of data for training.

## 4. Conclusions

HSI techniques can provide a real-time monitoring system to guide the precise application and reduction of pesticides and to provide objective and effective options for the automatic detection of crop damage caused by insect pests or diseases. In this research, we propose a deep learning classification and detection method that is based on band selection and a BEP that can be applied to determine the lowest cost to achieve the monitoring of leaf defects caused by RLF. To compensate for the deficiencies caused by band selection, the BEP method was selected to improve the detection efficiency. The results of the test dataset show that the use of the full-band classification is the best, and the band selection classification is better than the BEP. Except for criteria on skewness and signal-to-noise ratio, the accuracy of full-band classification is nearly 95%.

After using the trained model to predict the unknown samples, the results show that the CEM_band selection→DNN_band selection analysis method is the best model and has reached the expected prediction. The maximum DSC is 0.80, which means that its classification is 80%, which is identical to the classification recognized by entomologists. In addition, we discovered that the predictive area of the model was larger than the area observed by the human eye. This phenomenon may indicate that RLF damage may produce changes in parts of the spectrum that cannot be easily detected by the human eye. In addition, comparing the implementation of prediction operations based on the full-band DNN model and the band selection-based DNN model, the band selection method only needs 1% of the full-band time, which provides a vast potential for wider applications and has good rice identification capabilities. Only six bands are needed while reducing the technical cost required for on-site monitoring.

By providing more training data, the method also has significant room for improvement by implementing a data argumentation process or extending other data, such as the mean or variance-generating structures. While the current research has only been conducted in the laboratory or used non-specified multispectral images in the field, the handheld six-band sensor provided very good results, and its portability means that it could be adapted for use in the field to obtain realistic multispectral images on-site using

band Selection methods. In addition, most of the existing UAVs use CNN or vegetation indices for analysis and have not been studied much in spectral reflectance. As mentioned in Section 3.5, the HSI prediction model can detect infested areas before noticed by the human eye. This technique can be extended to UAV in the future to monitor the invisible spectral changes on the leaf surface. This technology can be extended to UAV in the future to monitor the invisible spectral changes on the leaves. Combining HSI techniques and deep learning classification models could provide real-time surveys that give on-site early warning of damage.

**Supplementary Materials:** The following are available online at https://www.mdpi.com/article/10.3390/rs13224587/s1, Figure S1: Bands selected through band prioritization and band decorrelation, Figure S2: Confusion Matrix result of DNN model, Figure S3: Prediction of spectral information from unknown rice sample, Figure S4: Entoscan Plant imaging system, Figure S5: Approximate time required for each step of the prediction of unknown samples, Table S1: Results of the first six bands of band selection using different criteria, Table S2: The accuracy of DNN classification evaluated by confusion matrix.

**Author Contributions:** Conceptualization, Y.-C.O. and S.-M.D.; methodology, Y.-C.O. and S.-M.D.; software, Y.-C.O.; validation, Y.-C.O. and S.-M.D.; formal analysis, G.-C.L.; investigation, G.-C.L.; resources, Y.-C.O. and S.-M.D.; data curation, G.-C.L.; writing—original draft preparation, G.-C.L.; writing—review and editing, Y.-C.O. and S.-M.D.; visualization, G.-C.L.; supervision, Y.-C.O. and S.-M.D.; funding acquisition, Y.-C.O. and S.-M.D. All authors have read and agreed to the published version of the manuscript.

**Funding:** This research was funded by the Ministry of Science and Technology (MOST), Taiwan (Grant No. MOST 107-2321-B-005-013, 108-23321-B-005-008, and 109-2321-B-005-024), and Council of Agriculture, Taiwan (Grant No. 110AS-8.3.2-ST-a6). The APC was funded by MOST 109-2321-B-005-024.

**Institutional Review Board Statement:** Not applicable.

**Informed Consent Statement:** Not applicable.

**Data Availability Statement:** Not applicable.

**Acknowledgments:** We are grateful to Chung-Ta Liao from Taichung District Agricultural Research and Extension Station for RLF collection and maintenance. We would also like to thank the publication subsidy from the Academic Research and Development of NCHU.

**Conflicts of Interest:** The authors declare no conflict of interest.

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
