# Peer review of "Detection and Classification of Rice Infestation with Rice Leaf Folder (Cnaphalocrocis medinalis) Using Hyperspectral Imaging Techniques"

_remotesensing, doi:10.3390/rs13224587_

Round 1

Reviewer 1 Report

The manuscript presents a study on the detection and classification of Cnaphalo-crocis medinalis damage in rice. The study has some significant weaknesses, as detailed below:
- English needs substantial work.
- Section 2.3.1: what are the spatial dimensions of the images?
- Table 1 is very difficult to understand. I had trouble inferring what the numbers mean. Also, tables and respective captions should be self-contained (independent from the main text).
- Lines 142-149: what vegetation index was used? Why not using one or more bands directly? This paragraph needs to be further detailed, because as is the text does not allow for proper reproduction.
- Lines 150-161: it is not clear how CEM was employed and how the objective. Did the authors try to isolate damaged tissue from the healthy regions? Is CEM really necessary here? Again, more details and a proper justification need to be provided.
- Section 2.7: it is not clear what the inputs for the deep learning models are. Again, a more comprehensive description is needed. Also, although the structures used in the network are typical of deep learning models, the architecture used is actually quite shallow. 
- Lines 258-259: since the samples used to test the model came from the same field and same variety of rice, you cannot say that the samples are completely independent, you can only say that they were different from those used for training.
- A serious weakness of the experiments is the fact that there is no direct comparison with other methods. The authors mention the accuracies reported in other studies, but since those were obtained on completely different datasets, the comparison is unreliable.
- The discussion section is very limited. There are many factors that can affect the results, such as severity of the damage, age of the leaf, presence of other stresses, rice variety, among others, but none of those are discussed. Also, nowhere in the text a proper justification for the complex approach adopted in the study is provided. It is possible that simply selecting the proper bands and applying deep learning directly to the background removed images would be enough to yield comparable results. The authors need to provide compelling arguments and experimental data to justify the proposed method.

Reviewer 2 Report

The authors propose a methodn for detecting rice infestation. The paper is on an interesting topic and the results are interesting and useful for the RS community. I list here below some comments to improve the presentation, but in general I would say that the manuscript requires minor revisions only.

LL.61-64
Reference is missing.
Could you offer more details on the previous studies (e.g. accuracies, what kind of data they used)?

2.7. Data Training Models
Why did you choose DNN as the classification algorithms?
Why did you set that configuration of hyperparameters?
Did you use the drop out?

Author Response

1. 61-64: Reference is missing. Could you offer more details on the previous studies (e.g., accuracies, what kind of data they used)?

Response: Thanks to the reviewer’s comments. The reference has been included in introduction. More information on previous studies were added as follows: “Fan et al. [21] applied a visible/near-infrared hyperspectral imaging system to detect early invasion of rice streak insects. Using the successive projection algorithm (SPA) [22], PCA and a back-propagation neural network (BPNN) [23] as classifiers to identify key wave-lengths, the classification accuracy of the calibration and prediction sets was 95.65%. Chen et al. [24] also employed a visible/near-infrared hyperspectral imaging system to acquire images and further developed a hyperspectral insect damage detection algorithm (HIDDA) to detect pests in green coffee beans. The method combines CEM and SVM and achieves 95% accuracy and a 90% kappa coefficient.”

2. Data Training Models

(1) Why did you choose DNN as the classification algorithms?

Response: Thanks to the reviewer’s comments. The main reason for using neural networks as classifiers in this study is that neural networks satisfy nonlinear multiple classifications at once. In addition, the development of neural networks is mature enough to be stable and fast as a classifier. The reason for using the DNN instead of the CNN is mentioned in Section 3.4 as follows: “The DNN multilayer perceptron model is suited for HSI data for classification because the spectral reflection of each pixel can form a vector”.

 (2) Why did you set that configuration of hyperparameters?

Response: Thanks to the reviewer’s comments. For future application to portable instruments, we wanted to perform predictive analysis in the fastest and easiest way. We conducted many pretests and discovered that the configuration of hyperparameters shown in Figure 4a is the best for obtaining stable results and the highest accuracy.

(3) Did you use the drop out?

Response: Thanks to the reviewer’s comments. We did not use the drop out since the training data were large enough and random sampling was performed during the training. Overfitting did not occur during the ten model tests, so dropout was not utilized.

Reviewer 3 Report

General comments: A most useful laboratory study on the assessment of damage by Rice Leaf Roller damage that could be could form the basis of methods adapted for use in the field.

Specific comments:

Title …rice infestation with rice leaf roller C…..

Line 13: ..early infestations cannot be detected visually. 

Line 21: I do not think it wise to use BS as an abbreviation as it has negative connotations of being not true.

…Then a band selection method was used to reduce…. Use “band selection” NOT BS throughout the paper.

Line 28: …and a Dice similarity coefficient of 0.8 which….

Line 35-36: …temperate countries [1] and has become one of the most important insect pests of rice cultivation.   Omit all reference to global warming: while it may be true this paper shows no evidence for it.

Line 49: …..inspect for damage visually…..

Line 50: …The uncertain timing of appearance of RLF means farmers are unable to predict pest arrival, so to avoid damage by undetected infestations, farmers often spray chemical pesticides preventively, leading to unnecessary…..

Line 56: …The external and internal damage caused by pest infestations, such as…..

Line 93: To obtain different levels of damage…

Figure 2: Explain what LWS, OP and MOP are since Figures should be able to be understood without having to refer to the materials and methods in the text.

Lines 121-129 need major revision: “The damage to leaves infested with different RLF instars (instar 1 to instar 5) for various durations of feeding (1-7 days) were assessed using VNIR hyperspectral imaging.  Leaves were placed flat on the conveyor belt to scan the image at every 90o turn to enlarge the data set.  Healthy leaves without RLF infestation were used as a control, and wilt on apex was also included to distinguish damage caused by RLF from non-RLF effects.  Before taking the VNIR hyperspectral images, light correction was conducted, and all processing of images was in a dark box to avoid interference from other light sources.  Information concerning the hyperspectral sensors and the dataset are shown in Table 1.”

Lines 138-139: where Rc is the hyperspectral image after calibration, Ro is the raw hyperspectral image, W is the standard white reference……

Line 168: Replace “Various BS methods” with Various band spectral methods     Also no BS in line 169.

Line 220: Please explain more fully what the “experts in agriculture” did as part of ground truth so we can better understand the basis of the ground truthing by the “entomologists” (lines 364-365) (are entomologists different from “experts in agriculture” of line 220?).  These ground truth assessments were a critical part of assessing the images in Figure 11.

Line 275-276: …used to calculate the similarity of two samples, in the value range between 0, where there is no overlap and 1 where there is complete overlap.

Line 292-293: There could be differences in spectral properties….. damage, which could be used as a basis for early identification of RLF infestations.

Line 296: Figure 8a.    and Line 299: Figure 8b.   and Line 302: in Figure 8a and 8b.

Line 299: …healthy leaves (D0)…

Line 303: … contain a lot of redundant information that will slow analysis…. 

Line 334: Again in the Figures, please state abbreviations: state what ROI, CEM, RGB are.

Lines 327, 330 etc: use damage NOT damages; AND please use Band Selection NOT BS can reach the best performance in ROI detection….

Line 370: rice leaves NOT leave

Lines 375-385: Far too many abbreviations make these section difficult to follow.  I suggest something like: “Similar to the results in Figure 11, analysis methods showed that CEM_BS>DNN_BS showed the best performance (Table 3), because this method showed the highest TP (correct identification of OP) and Overall Accuracy (OA) and lowest FN (misidentification of OP).  However very high False Positives (FP) were obtained from the methods of CEM_Full-band……  then continue lines 381-ff

Line 399-400: ..as the best model for predicting unknown samples as it had the highest accuracy, recall, DSC and took the shortest time.  Although the analysis……also showed a reasonably good performance,..

Line 411-412: are a few words missing?

Line 418: Automatic detection of plant pests is extremely useful as it reduces the tedious work of monitoring large paddy fields, and it will detect…

Line 433: by the naked eye. 

Line 461: This phenomenon may indicate that RLF damage may result in changes in parts of the spectrum that cannot be easily detected by the human eye.  

The testing of this method in the field is a critical next step but having a hand-held 6-band sensor means that this method could be adapted for use in the field. So emphasize this by changing lines 469-ff to something like:

“While the current research has only been conducted in the laboratory, the hand-held 6-band sensor provided very good results and its portability means that it could be adapted for use in the field to obtain realistic multi-spectral images on-site using Band Selection methods. Combining HSI techniques and deep learning classification models could provide real time surveys that give on-site early warning of damage.”

Reviewer 4 Report

Overal this is a good paper in terms of the content and the application. What it needs is some editing of the English which detracts from the content. It would be nice of some of the paper was a little wordy, and some sentences could be removed from some places and placed elsewhere in the paper. The illustrations are good. The logic of the paper is clearly executed. The discussion and conclusions are a little limited and I would to see more on the practicality of such an approach which while clearly showing potential but despite mention of commercialization possibilities, the study is very much lab based and some indication of how this could be applied in practice would be good.   To begin with the paper needs the help of a professional editor to tidy up the communication plus some additions to the content.

Author Response

Overall, this is a good paper in terms of the content and the application. What it needs is some editing of the English which detracts from the content. It would be nice of some of the paper was a little wordy, and some sentences could be removed from some places and placed elsewhere in the paper. The illustrations are good. The logic of the paper is clearly executed. The discussion and conclusions are a little limited and I would like to see more on the practicality of such an approach which while clearly showing potential but despite mention of commercialization possibilities, the study is very much lab based and some indication of how this could be applied in practice would be good. To begin with the paper needs the help of a professional editor to tidy up the communication plus some additions to the content.

Response: Thank the reviewers for their valuable comments. We have sent the manuscript to a manuscript editing service for a review of the English in the paper and will submit the corresponding certificate with the manuscript. In future applications, the key wavelengths identified through the band selection of this study can be applied to the multispectral sensor of UAVs and portable instruments. Our ongoing studies have applied the key wavelengths on a custom-made, multispectral camera for detecting the infestation of RLF. The results are quite promising.

Round 2

Reviewer 1 Report

The authors provided some adequate answers to my concerns, but I am still not convinced that the proposed method is advantageous. The lack of direct comparison with other methods and the absence of a proper analysis of the factors that can decrease the effectiveness of the algorithm are particularly troublesome.

Author Response

Thanks for the reviewer’s comment. We added some comparison with other methods into the “3.6 Discussion” as follows: “Traditional classifiers such as linear SVM (support vector machine) and LR (logistic regression) can be attributed to single-layer classifiers, while decision trees or SVM with kernels are considered to have two layers [40,41]. However, deep neural architectures with more layers can potentially extract abstract and invariant features for better image or signal classification [42]. Our previous studies to detect Fusarium wilt on Phalaenopsis have shown this result [43]. In addition, we have used the Entoscan Plant imaging system to detect the infestation of RLF, but this system only covers 16 bands (390, 410, 450, 475, 520, 560, 625, 650, 730, 770, 840, 860, 880, 900, 930 and 960 nm) to obtain the Normalized Difference Vegetation Index. It may not be specific enough to distinguish the damaged caused by different pests. Therefore, we attempt to find a more representative vector from the spectral fingerprint of the hyperspectral imaging system to detect the infestation of RLF. At the same time, the band selection was used to remove redundant information to achieve the time required for the automatic detection process. It is not only reducing the time by 2.45 times (from 8'11'' to 3'20"), but also reach higher accuracy (0.981) than that (0.951) in the full band. The six bands (489, 501, 603, 664, 684, and 705 nm) obtained through band selection are more representative than bands supplied by the Entoscan Plant imaging system and can be applied to the multispectral sensor of UAVs and portable instruments for field use”.

Reviewer 4 Report

It would appear that the authors have addressed the issues raised by the reviewers and have attended to the English language etc. However, I do not think they have expanded the paper to cover a little more context about the value of this theoretical/lab-based approach to the 'real-world' and I think placing the paper in a wider context/future developments would add value to the paper. At present the changes made are the easy ones that have been pointed out...... rather than perhaps looking at the paper and its coverage a little more.

Author Response

Thank you so much for your comment. Extended background/future developments have been added to the “3.6 Discussion” and “4. Conclusions” of revised manuscript.

  1. In the end of “6 Discussion”, the following sentences were added: “The methods, algorithms, and models we established in this paper will be applied to other important rice insect pests and verify in the field by using either UAVs or portable instruments that carried the multispectral sensor. In addition, a platform to integrate all these information will be established to interact with farmers”.
  2. In the “4. Conclusions”, the following paragraph were added to replaced the original one: While the current research has only been conducted in the laboratory or used non-specified multi-spectral images in the field, the hand-held 6-band sensor provided very good results and its portability means that it could be adapted for use in the field to obtain realistic multi-spectral images on-site using band Selection methods. In addition, most of the existing UAVs use CNN or vegetation indices for analysis and have not been studied much in spectral reflectance. As mentioned in Section 3.5, HSI prediction model can detect infested areas before noticed by the human eye. This technique can be extended to UAV in the future to monitor the invisible spectral changes on the leaf surface. This technology can be extended to UAV in the future to monitor the invisible spectral changes on the leaves. Combining HSI techniques and deep learning classification models could provide real-time surveys that give on-site early warning of damage.
